# Exploring the Contribution of the AcrB Homolog MdtF to Drug Resistance and Dye Efflux in a Multidrug Resistant *E. coli* Isolate

**DOI:** 10.3390/antibiotics10050503

**Published:** 2021-04-28

**Authors:** Sabine Schuster, Martina Vavra, Ludwig Greim, Winfried V. Kern

**Affiliations:** 1Division of Infectious Diseases, Department of Medicine II, University Hospital and Medical Center, 79106 Freiburg, Germany; martina.vavra@uniklinik-freiburg.de (M.V.); ludwig.greim@med.uni-duesseldorf.de (L.G.); winfried.kern@uniklinik-freiburg.de (W.V.K.); 2Faculty of Medicine, Albert-Ludwigs-University, 79106 Freiburg, Germany

**Keywords:** MdtF (YhiV), multidrug resistance, RND-type efflux pump, dye accumulation, real-time efflux

## Abstract

In *Escherichia coli*, the role of RND-type drug transporters other than the major efflux pump AcrB has largely remained undeciphered (particularly in multidrug resistant pathogens), because genetic engineering in such isolates is challenging. The present study aimed to explore the capability of the AcrB homolog MdtF to contribute to the extrusion of noxious compounds and to multidrug resistance in an *E. coli* clinical isolate with demonstrated expression of this efflux pump. An *mdtF*/*acrB* double-knockout was engineered, and susceptibility changes with drugs from various classes were determined in comparison to the parental strain and its *acrB* and *tolC* single-knockout mutants. The potential of MdtF to participate in the export of agents with different physicochemical properties was additionally assessed using accumulation and real-time efflux assays with several fluorescent dyes. The results show that there was limited impact to the multidrug resistant phenotype in the tested *E. coli* strain, while the RND-type transporter remarkably contributes to the efflux of all tested dyes. This should be considered when evaluating the efflux phenotype of clinical isolates via dye accumulation assays. Furthermore, the promiscuity of MdtF should be taken into account when developing new antibiotic agents.

## 1. Introduction

Knowledge about resistance mechanisms is crucial for future drug development in particular against Gram-negative pathogens. Many of them rank within the uppermost levels of the WHO priority list for the urgent requirement for new antibiotics [1]. The importance of RND (resistance nodulation cell division)-type efflux pumps for the emergence of multidrug resistance (MDR) in Gram-negative pathogens has been demonstrated, among others, for *Escherichia coli* [2,3,4], *Klebsiella* [5], *Enterobacter* [6] *Acinetobacter* [7], and *Pseudomonas aeruginosa* strains [4]. As shown recently with the latter, in isolates lacking carbapenemases, the overexpression of efflux pumps appeared as the predominant reason for carbapenem and multidrug resistance [8]. In contrast to *Pseudomonas aeruginosa* strains (in which several different transporters of the RND superfamily contributes to MDR), AcrB has been the only multidrug efflux transporter with a proven impact in *E. coli*. The relevance of further efflux pumps encoded in the *E. coli* chromosome has been less explored. One reason is that they are not found significantly expressed in laboratory strains [9] (in which most of the experiments were carried out), whereas results from MDR clinical isolates have remained underrepresented.

In recent studies with such isolates, a significant percentage of the collection revealed *mdtF* expression in addition to *acrB* expression [10,11]. MdtF (formerly YhiV), exhibiting a sequence similarity of 79% compared with AcrB [12], is complexed with the membrane fusion protein MdtE (formerly YhiU) and with TolC, the ubiquitous outer membrane channel in *E. coli* not only working together with RND-type efflux pumps but also with transporters of other families, such as the ABC-transporter MacB and the MFS-transporter EmrB [13]. The principal capability of MdtF to efflux drugs from different classes, with the exception of the oxazolidinone linezolid, has been shown [9], but little is known about a potential role in MDR development. In the present study, we aimed to explore the MdtF functionality and its contribution to drug resistance in an MDR *E. coli* isolate with verified expression of this RND-type efflux pump [3,11].

## 2. Results and Discussion

### 2.1. Impact of mdtF Inactivation on Drug Susceptibility

In order to evaluate the contribution of MdtF to total TolC-dependent efflux in MDR *E. coli* strains, we aimed to compare an *mdtF*/*acrB* double-knockout of the patient isolate KUN9180 (besides *acrB* also showing *mdtF* expression) with the *acrB* (KUN∆*acrB*) and *tolC* single-knockout (KUN∆*tolC*) mutants from previous studies [3,11]. Because selection options are limited in MDR strains, the reutilization of the kanamycin/neomycin selection cassette used for switching off *acrB* was an option. Thus, the first step was its removal from *acrB* and the second step was its reintroduction in *mdtF* (see Section 3.3). The resulting double-knockout was subjected to susceptibility testing, and the results were compared with those from the KUN∆*acrB* and KUN∆*tolC* single-knockout mutants. Between the latter, significant differences in the susceptibilities to nadifloxacin, zoliflodacin [11], and novobiocin (Table 1) had been shown, but not to several other proven AcrB substrates, including the more hydrophilic fluoroquinolones levofloxacin and moxifloxacin and the non-fluoroquinolone gyrase inhibitor gepotidacin [11], to tetracycline, chloramphenicol, linezolid, clindamycin, and rifaximin (Appendix A), all of which were demonstrably effluxed in isolate KUN9180 [3,11]. Regarding those results, only nadifloxacin, zoliflodacin, and novobiocin are potentially extruded from a TolC-dependent transporter other than AcrB.

Indeed, we detected susceptibility increases for these three agents with the KUN∆*acrB*∆*mdtF* double-mutant in comparison with the *acrB* single-knockout, but the changes were small, with significance only confirmed for nadifloxacin (*p* value 0.01). Remarkably, the MICs remain one or two dilution steps above those shown for the *tolC* deletion mutant (Table 1). This could be due to another TolC-dependent efflux pump. However, we did not find any significant expression of such an additional transporter in the KUN9180 isolate [11]. Residual differences in the susceptibilities of the ∆*acrB*∆*mdtF* and the *tolC* inactivated mutants could be explained by further physiological functions of TolC [13]. Among others, a role in enterobactin export has been reported [14]. An increasing accumulation of the siderophore in the periplasm due to *tolC* deletion could impair the bacterial fitness and thereby, to some extent, drug susceptibilities [15]. We have to outline that we did not include investigations of a putative physiological impact of MdtF under differing environmental conditions other than drug exposure in this study. A previous report had detected a protective role of MdtF against nitrosative damage under anaerobic growth conditions [16].

### 2.2. Impact of mdtF Inactivation on Intracellular Dye Accumulation

To gain further insights regarding the functionality and specificity of the MdtF transporter, we carried out accumulation assays with six chemically diverse fluorescent dyes which are known substrates of AcrB. In a previous study with the MDR clinical isolate KUN9180, we had observed residual efflux of dyes not fully prevented after AcrB inactivation. That was in contrast to findings with derivatives of laboratory *E. coli* strains, such as the *acrB* overexpressing K-12 derivative 3-AG100 [2,3]. A recent study with new powerful pyranopyridine efflux pump inhibitors showed an increase in the accumulation of the fluorescent dye Hoechst 33342 exceeding that caused by *acrB* inactivation, which also suggests the existence of at least one more active efflux transporter [17]. 

The accumulation assays with the MDR *E. coli* isolate reveal a remarkable contribution of MdtF to the total efflux of all dyes tested, because only the *acrB*/*mdtF* double-knockout (but not KUN∆*acrB*) reached the intracellular accumulation comparable with that of the *tolC* knockout (Figure 1). Moreover, most of these dyes seemed to be even better substrates of MdtF than of AcrB. The latter contributes most efficiently to total TolC-dependent efflux only in the case of Pyronine Y, whereas all other dyes appeared more successfully expelled by MdtF. 

Regarding berberine and β-naphthylamine, the latter being the intracellular fluorescent cleavage product of the efflux pump inhibitor PAβN (phenylalanine arginine β-naphthylamide) [18], the accumulation seemed even higher within the *acrB*/*mdtF* double-knockout versus the KUN∆*tolC* mutant, but the differences were not significant (*p* values > 0.05).

The results may help to resolve the previous discussion about ethidium as a substrate of AcrB in different *E. coli* isolates [19]. In some *E. coli* strains (including the clinical strain investigated in the present study), the inactivation of *acrB* does not remarkably affect the efflux of ethidium. We here show that this is associated with the presence of MdtF which obviously strongly reduced ethidium accumulation in a ∆*acrB* background (Figure 1).

### 2.3. Impact of mdtF Inactivation on Real-Time Efflux

Since intracellular compound accumulation might also be impaired by altered influx or uptake, we conducted real-time efflux assays with three different dyes suitable for this application. We used Nile red [20], 1,2′-dinaphthylamine (1,2′-DNA) [21], and the piperazine arylideneimidazolone BM-27 [22] to assess the extrudability of these substances in the KUN9180 clinical isolate and its mutants. Obviously, MdtF remarkably contributes to the efflux of these dyes, too (Figure 2). Moreover, it appears as major efflux transporter in the case of BM-27. This is unlike the findings with the *acrB* overexpressing K-12 derivative 3-AG100, in which the inactivation of *acrB* caused an efflux breakdown reaching the level of the *tolC*-knockout for all dyes tested [11], but in accordance with the fact that *mdtF* is not expressed in 3-AG100 and its ∆*acrB* derivative [3]. Some residual efflux with Nile red and possibly also with 1,2′-DNA occurred in the *acrB*/*mdtF* double-knockout mutant (Figure 2) suggesting the putative activity of another TolC-dependent transporter. However, as just mentioned, we could not detect any significant expression of such an efflux pump besides AcrB and MdtF in the KUN9180 isolate [11]. 

Differences in the substrate compatibility with MdtF regarding the dyes used in the assays are probably due to different physicochemical properties of these compounds. They all are highly lipophilic, but differ remarkably for instance in size and shape (Appendix A). BM-27 is the largest molecule with a molecular weight of 425 [22], whereas that of Nile red is 318 and that of 1,2′-DNA is only 269. (https://pubchem.ncbi.nlm.nih.gov/compound/; accessed on 30 March 2021) [23]. Of course, many other determinants (such as the polarity of the molecules or available hydrogen bond donors and acceptors) might play a role.

## 3. Materials and Methods

### 3.1. Strains, Growth Conditions, Chemicals

The MDR clinical isolate KUN9180 was kindly provided from Yasufumi Matsumura from the Department of Clinical Laboratory Medicine, Kyoto University Graduate School of Medicine, Japan. The *acrB* and the *tolC* knockout mutants of the MDR *E. coli* isolate KUN9180 had been engineered within the scope of previous studies [3,11]. The KUN∆*acrB*∆*mdtF* double-knockout was constructed as described in Section 3.3. All strains and mutants were cultured in cation-adjusted Mueller Hinton broth (MH2) or on MH2 agar plates at 37 °C overnight.

Chemicals were purchased from Merck (Darmstadt, Germany) with the exception of 1,2′-DNA, which was purchased from TCI (Eschborn, Germany), and BM-27, which was a kind gift from Jadwiga Handzlik from the Department of Technology and Biotechnology of Drugs, Jagiellonian University Medical College, Faculty of Pharmacy, (Kraków; Poland).

### 3.2. Susceptibility Testing 

Minimal inhibitory concentrations (MICs) of antibiotics were determined by broth microdilution assays according to EUCAST guidelines (https://www.eucast.org/; accessed on 30 March 2021). 

### 3.3. Engineering of Mutant KUN∆acrB∆mdtF 

Mutant KUN∆*acrB* (KUN9180Δ*acrB*:FRT-PGK-gb2-neo-FRT [3]) was used to engineer the double-mutant KUN∆*acrB*∆*mdtF*. In the first step, the FRT-PGK-gb2-neo-FRT cassette was removed from *acrB* with the assistance of an FLPe (708-FLPe) expression plasmid (Gene Bridges, Heidelberg, Germany), as described in the manual of the “Quick & Easy *E. coli* Gene Deletion Kit” (Gene Bridges). The resulting *acrB* deficient mutant without any resistance marker was cured from the FLPe plasmid (with a temperature sensitive origin) by cultivating at 37 °C and then transformed with a curable Red/ET plasmid (Gene Bridges) harboring a chloramphenicol resistance marker. An FRT-PGK-gb2-neo-FRT cassette was PCR-amplified using oligonucleotides with flanks homologous to the desired substitution region in *mdtF* (Table 2). Red/ET recombination with the purified PCR product was carried out as described in the “Quick & Easy *E. coli* Gene Deletion Kit” protocol (Gene Bridges). Recombinants were selected on agar plates containing 100 µg/ml kanamycin (cross-resistance with neomycin), and successful insertion of the FRT-PGK-gb2-neo-FRT cassette was verified by PCR with check-primers binding up- and downstream from the replacement site in *mdtF* (Table 2).

### 3.4. Dye Accumulation Assays

The intracellular accumulation of fluorescent dyes was determined as described previously [24,25,26], with minor modifications. Briefly, bacteria from an overnight cultivated MH2 agar plate were suspended in PBS (phosphate buffered saline, pH 7.4) supplemented with MgCl_2_ and with glucose to final concentrations of 1 mM and 0.4%, respectively. The suspensions were adjusted to an OD_600_ of 1 and immediately used for determining the intracellular accumulation of dyes, which were added to a final concentration of 2.5 µM for ethidium bromide, 2.5 µM for Hoechst 33342, 30 µg/mL for berberine, 200 µM for PAβN (β-napththylamine determination), and 22.5 µM for Pyronine Y. Fluorescence was measured in the TECAN Infinite M200 PRO plate reader (Crailsheim, Germany) with an incubation temperature of 37 °C. For ethidium, the excitation and emission wavelengths were 518 and 605 nm, respectively, for Hoechst 33343 350 and 461 nm, for berberine 355 and 517 nm, for β-naphthylamine 320 and 460 nm, and for Pyronine Y 545 and 570 nm. Fluorescence values were corrected by subtracting the values detected from the bacterial suspensions without dye.

### 3.5. Real-Time Efflux Assays

Real-time efflux assays were conducted with Nile red, 1,2′-DNA, and BM-27 according to protocols published earlier [20,21,22] with slight modifications. Briefly, 20 mL MH2 broth was inoculated with a colony from a freshly grown MH2 agar plate and cultivated overnight at 37 °C with shaking (200 rpm). Bacterial cells were harvested by centrifugation (3220 g, 10 min) and washed twice with PBS. The pellets were suspended in PBS containing 1 mM MgCl_2_ and adjusted to an OD_600_ of 1. Efflux arrest was induced by incubating with the proton gradient uncoupling agent CCCP (carbonyl cyanide 3-chlorophenylhydrazone, final concentration 5 µM) at 37 °C for 20 min followed by dye loading with Nile red or BM-27 to final concentrations of 10 µM at 37 °C for 2 h. 1,2′-DNA was added to a final concentration of 32 µM (4 h incubation, 37 °C). To monitor the real-time efflux, 180 µL aliquots of cells were washed by centrifugation (5800 g, 2 min) and resuspended in the same volume of PBS containing 1 mM MgCl_2_. After 40 s of fluorescence recording with the TECAN plate reader M200 Pro, cells were re-energized by the addition of glucose to a final concentration of 1 mM and the measurement was continued for 360 s (at 37 °C). For Nile red the excitation and emission wavelengths were 544 and 650 nm, respectively, for BM-27 400 and 457 nm, and for 1,2′-DNA 370 and 420 nm.

### 3.6. Statistical Data Analysis

Standard deviations (SD) were calculated from the mean of *n* experiments as indicated and the statistical significance of differences was analyzed by two-tailed t-tests using the software GraphPad Prism (San Diego, CA, USA) version 8.4.2 (*p* values < 0.05 represent significance).

## 4. Conclusions

We found limited contribution of the RND-type transporter MdtF to the antibiotic resistance profile of an MDR *E. coli* isolate, but a remarkable capacity to export dyes (including ethidium) from different chemical substance classes suggesting a potential risk that new compounds including drugs could be substrates, too. Furthermore, it should be kept in mind that dye assays established to evaluate the efflux competence of bacteria do not necessarily allow conclusions about the activity of the major drug exporter AcrB in MDR *E. coli* isolates.

## Figures and Tables

**Figure 1 antibiotics-10-00503-f001:**
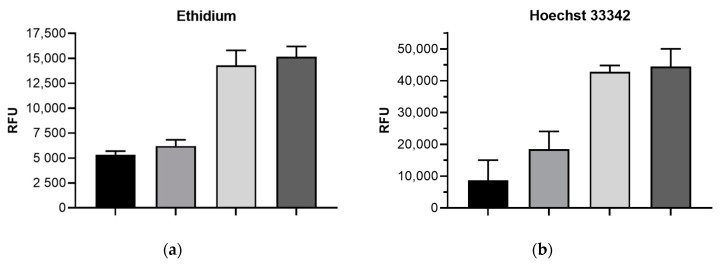
Accumulation of dyes in isolate KUN9180 and derived mutants (RFU, relative fluorescence unit; bars represent means of ≥3 assays ±SD from values at 30 min, for Pyronine Y at 20 min of incubation): (**a**) ethidium; (**b**) Hoechst 33342; (**c**) berberine; (**d**) β-naphthylamine; (**e**) quenching of fluorescence by intracellular accumulated Pyronine Y (lower fluorescence values reflect higher accumulation of Pyronine Y). (**f**) Graph legend for panels (**a**–**e**).

**Figure 2 antibiotics-10-00503-f002:**
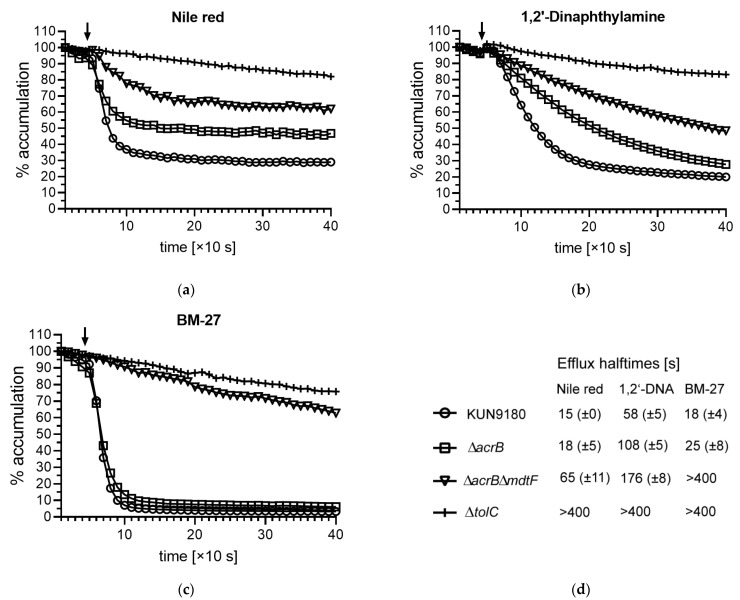
Real-time efflux with fluorescent dyes (the arrow indicates the addition of glucose for the re-energization of the efflux-arrested cells loaded with dye): (**a**) Nile red; (**b**) 1,2’-dinaphthylamine (1,2′-DNA); (**c**) BM-27. (**d**) Graph legend for panels (**a**–**c**) and efflux halftimes (means with SD, *n* ≥ 3) detected with the KUN9180 isolate and its mutants.

**Table 1 antibiotics-10-00503-t001:** Drug susceptibilities for which significant difference between *acrB* and *tolC* mutants of the MDR *E. coli* isolate KUN9180 were reported.

	MIC in µg/mL ^1^		
*E. coli* Strain/Mutant	Nadifloxacin	Zoliflodacin	Novobiocin
KUN9180	>512 (0)	4 (0)	128 (64)
KUN∆*acrB*	32 (16)	0.25 (0)	4 (2)
KUN∆*acrB*∆*mdtF*	16 (0)	0.125 (0)	2 (0)
KUN∆*tolC*	4 (2)	0.045 (0.02)	1 (0)

^1^ MIC, minimal inhibitory concentration; the median of ≥7 independent assays is shown and the MAD (median absolute deviation) is given in parenthesis.

**Table 2 antibiotics-10-00503-t002:** Oligonucleotides used in this study.

Oligonucleotide	Sequence (5′-3′) ^1^	Application
upOl-*mdtF*-FRT-PGKgb2neo	*gtcactcaggtgattgagcaaaatatgaatgggcttgatggcctgatgta* aattaaccctcactaaagggcg	FRT-PGK-gb2neo-FRTcassette amplification
lowOl-*mdtF*-FRT-PGKgb2neo	*gcggtgccatcgtgccagaggcgttgcgtacataccattggttgatgtta* taatacgactcactatagggctc	FRT-PGK-gb2neo-FRTcassette amplification
Check-*mdtF*-fw	ggcgatcatgaacttaccgg	Check-primer for *mdtF* insertions
Check-*mdtF*-rv	ggatgccgttgtagcgttc	Check-primer for *mdtF* insertions

^1^ Underlined letters represent the primer sequence for the FRT-PGKgb2neo-FRT template, letters in italic the homology flanks fitting to *mdtF*.

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
