# Peer review of "Exploring the Contribution of the AcrB Homolog MdtF to Drug Resistance and Dye Efflux in a Multidrug Resistant E. coli Isolate"

_antibiotics, 2021, doi:10.3390/antibiotics10050503_

Round 1
Reviewer 1 Report
I commend the authors on an interesting work. The elucidation of resistance mechanisms is of utmost importance in Gram-negative bacteria.
Abstract:
L13: to the multidrug resistant phenotype in an E. coli
Introduction:
L29-L35: please discuss that overexpressed efflux pumps may lead to uncommon phenotypes in ESKAPE pathogens, which may not be explained by other kinds of resistance mechanisms:
https://www.ncbi.nlm.nih.gov/pmc/articles/PMC7235726/
L56: Because/As selection options…
- Results and discussion:
I think the authors should make more of an effort to discuss their findings in the context of other literature results, as currently, it is heavily focused on the „Results” portion.
Methods:
the methodologies used were described adequately
I suggest that the authors put in a separate figure with the efflux pump inhibitors and dyes used in the assays for the readers to easily access them.
Author Response
- L13 (abstract): We replaced “While there was limited impact on the susceptibility to drugs, the RND-type transporter remarkably contributes to the efflux of all tested dyes” by “The results show that there was limited impact to the multidrug resistant phenotype in the tested coli strain, while the RND-type transporter remarkably contributes to the efflux of all tested dyes”
- L29-L35 (introduction): In L30, we have included a passage outlining the role of overexpressed efflux pumps that can explain the carbapeneme resistance phenotype in isolates lacking carbapenemases and also the appropriate literature.
- L56: We replaced “Since” by “Because”.
- Results and discussion: We now also mention a report of the physiological role of MdtF under anaerobic conditions (L84). There is not extensive literature regarding MdtF available, but we tried to include the most important aspects in the manuscript which is designated to be published as a “Communication”.
- Methods: We additionally prepared a figure (Figure S1) with the structures of the dyes and the inhibitor phenylalanine arginine beta-naphthylamide used in this study and included it in the “Supplementary Material” (Figure S1). An indication of the newly added supplement is now given in the manuscript (following the “Conclusion” section). With regard to the properties of compound BM-27 (L146-L148 in the initial manuscript), we have included a small correction, because there was a confusion with the compound. The reason is that the BM-27 found in PubChem is not identical with the piperazine arylideneimidazolone BM-27 from Jadwiga Handzlik (reference is given, it is not provided in PubChem) with the consequnence that for this compound the polar surface area is not available, only molecular weight and solubility. So we could only compare these properties.
Reviewer 2 Report
This manuscript by Schuster and colleagues explores the capacity of MdtF, a homologue of the RND family AcrB protein in E. coli, to efflux cytotoxic compounds in a TolC-dependent fashion from a clinical isolate that produces the protein. The authors constructed and compared an mdtF/acrB double-knockout of patient isolate KUN9180 with the acrB and tolC single-knockout mutants of the same isolate to determine the substrate efflux profile of MdtF-TolC. A suite of well-established experimental procedures for assessing efflux (whole cell dye accumulation and efflux assays that exploited fluorescence properties of the substrates tested, and drug susceptibility assays) were performed. The experimental approach is typical for this type of work; the experimental methodologies are clearly described, and the experiments performed in a competent manner and with appropriate controls in place. Statistical analysis and presentation of the data was appropriate. The conclusions drawn from the presented data are perfectly reasonable. In general, the paper is very nicely and concisely written, with excellent use of English throughout. Referencing is also appropriate throughout. In summary, the presented work adds to our knowledge of bacterial multidrug efflux proteins.
I have only one point that the authors should address: on line 66 the authors state that they have not presented data for a particular set of experiments (“data not shown”). The data referred to should be included as ‘Supplementary Material’.
Author Response
L66 (L64-L68): We included a table with the MIC values of the mentioned drugs (determined with the mutants and the parental E. coli isolate) in the newly added “Supplementary Material” (Table S1). In L64-L68 ( initial manuscript), we specified more precisely the drugs which were additionally shown in Table S1.